# Homeostatic regulation through strengthening of neuronal network-correlated synaptic inputs

**Samuel J Barnes[1,2], Georg B Keller[3], Tara Keck[4]***

[1]Department of Brain Sciences, Division of Neuroscience, Imperial College London, Hammersmith Hospital Campus, London, United Kingdom; [2]UK Dementia Research Institute at Imperial College, London, United Kingdom; [3]Friedrich Miescher Institute for Biomedical Research, Basel, Switzerland; [4]Department of Neuroscience, Physiology and Pharmacology, University College London, London, United Kingdom

**Abstract** Homeostatic regulation is essential for stable neuronal function. Several synaptic mechanisms of homeostatic plasticity have been described, but the functional properties of synapses involved in homeostasis are unknown. We used longitudinal two-photon functional imaging of dendritic spine calcium signals in visual and retrosplenial cortices of awake adult mice to quantify the sensory deprivation-induced changes in the responses of functionally identified spines. We found that spines whose activity selectively correlated with intrinsic network activity underwent tumor necrosis factor alpha (TNF-α)-dependent homeostatic increases in their response amplitudes, but spines identified as responsive to sensory stimulation did not. We observed an increase in the global sensory-evoked responses following sensory deprivation, despite the fact that the identified sensory inputs did not strengthen. Instead, global sensory-evoked responses correlated with the strength of network-correlated inputs. Our results suggest that homeostatic regulation of global responses is mediated through changes to intrinsic network-correlated inputs rather than changes to identified sensory inputs thought to drive sensory processing.

*For correspondence:
t.keck@ucl.ac.uk

**Competing interest:** The authors declare that no competing interests exist.

## Editor's evaluation

When sensory inputs, such as vision or sound, are chronically disabled, the loss of input activity is counterbalanced by the upregulation of synaptic activity. In this study, the authors have addressed homeostatic responses in adult animals and provide evidence that instead of synapses that directly represent the sensory information, synapses that show correlated intrinsic network activity are the ones that undergo the change upon sensory deprivation. This fundamental and important paper whose key claims are well supported by the data, will be useful to readers in the fields of experience-dependent plasticity, sensory cortical coding, and homeostatic plasticity.

## Introduction

Homeostatic regulation of neuronal firing levels is a critical feature of neural circuits. It allows for prolonged stable function during changing activity levels associated with ongoing Hebbian plasticity, changes in sensory drive or input loss (*Keck et al., 2017*). The impairment of homeostatic regulation is thought to result in prolonged periods of excessive neuronal hyper- or hypoactivity, which may impair network function (*Frere and Slutsky, 2018*; *Litwin-Kumar and Doiron, 2014*; *Marder and Prinz, 2002*; *Tetzlaff et al., 2011*; *Wu et al., 2020*; *Zenke et al., 2013*). Homeostatic regulation occurs through plastic changes to a broad range of synaptic, intrinsic, cellular, and

network-level properties (*Barnes et al., 2015*; *Bridi et al., 2020*; *Bridi et al., 2018*; *Gainey and Feldman, 2017*; *Joseph and Turrigiano, 2017*; *Keck et al., 2017*; *Lee and Kirkwood, 2019*; *Ma et al., 2019*; *Turrigiano, 2011*). These changes must facilitate rebalancing of activity while maintaining the underlying circuits established for ongoing sensory processing (*Chen et al., 2013a*; *Iacaruso et al., 2017*), as well as for learned (*Jurjut et al., 2017*; *Lee et al., 2020*; *Poort et al., 2015*) and innate behaviors (*Branco and Redgrave, 2020*; *Evans et al., 2018*). Maintaining this balance is particularly challenging with synaptic forms of homeostatic plasticity, where synaptic connections form the basis for circuitry and memory, but may also undergo homeostatic changes in strength that is inversely proportional to changes in activity levels. Previous work in adult animals has shown that synaptic homeostatic changes occur in only a subset of synapses (*Barnes et al., 2017*) or via additive changes (*Goel and Lee, 2007*), as opposed to multiplicative changes across the entire population described for some homeostatic mechanisms during development and the critical period (*Desai et al., 2002*; *Kaneko et al., 2008*; *Turrigiano, 2017*; *Turrigiano et al., 1998*). Identifying the functional properties of the subset of synapses that are homeostatically strengthened in adulthood is essential for understanding how sensory processing and homeostatic regulation are coordinated after the closure of the critical period, when sensory plasticity is reduced (*Hensch, 2005*; *Levelt and Hübener, 2012*).

Sensory cortices in adult mice are known to be multimodal and receive multiple sensory inputs (*Ibrahim et al., 2016*; *Iurilli et al., 2012*; *Knöpfel et al., 2019*; *Todd et al., 2016*; *Vann et al., 2009*; *Yoshitake et al., 2013*), as well as intrinsic contextual and intracortical inputs (*Fiser et al., 2016*; *Khan and Hofer, 2018*; *Miller et al., 2014*; *Okun et al., 2015*; *Pakan et al., 2016*; *Pakan et al., 2018b*; *Petrus et al., 2015*; *Roth et al., 2016*; *Saleem et al., 2018*). Large populations of neurons in visual cortex have a strong correlation or coupling with the average network activity, which is thought to be largely non-sensory and represent contextual input that may be associated with behavioral state (*Okun et al., 2015*; *Stringer et al., 2019*). High levels of network coupling are correlated with a higher probability of local intracortical connections between neurons (*Okun et al., 2015*) and could also reflect the presence of other common non-sensory inputs (*Yoshimura et al., 2005*). Following sensory deprivation, functional strengthening could occur in functional subsets of inputs, such as the remaining sensory inputs, other non-deprived sensory inputs, or network-correlated inputs, either independently or in a combination of subsets; however, it is currently unclear whether the synapses that undergo homeostatic strengthening encode a specific functional input.

Here, we examine the properties of inputs that undergo functional changes following sensory deprivation using repeated two-photon functional imaging of dendritic spines in awake mice (*Chen et al., 2013b*; *Iacaruso et al., 2017*). We find that, following deprivation, inputs with activity that is correlated with the network activity undergo functional strengthening, while inputs that have identified sensory-evoked responses do not strengthen, independent of whether they encode a deprived or non-deprived sensory input. Despite the fact that sensory-evoked spine responses are not strengthened, we find that global sensory-evoked responses homeostatically increase following deprivation and are correlated with increases in responses of network-correlated spines on their dendrites. Together, our work suggests that homeostatic synaptic regulation occurs through changes to network-correlated inputs, rather than the sensory responsive inputs that drive ongoing sensory processing.

## Results

We used chronic two-photon imaging in awake adult mice expressing the genetically encoded calcium indicator GCaMP6s (see Methods) to examine the functional properties of dendritic spines that undergo functional plasticity following sensory deprivation. Over a span of 4 days, we repeatedly measured functional calcium signals in dendritic spines on the distal tufts (in layer 2/3) of layer 5 pyramidal cells in primary monocular visual cortex before and after sensory deprivation via enucleation of the contralateral eye (*Figure 1A–C*; *Barnes et al., 2015*). We extracted spine responses using previously published methods (*Chen et al., 2013b*; *Iacaruso et al., 2017*; *Wilson et al., 2016*) and were able to measure the functional response properties, response amplitude, and response frequency of the same spines over the course of the imaging time series (*Figure 1B–C*, *Figure 1—figure supplement 1A–D*).

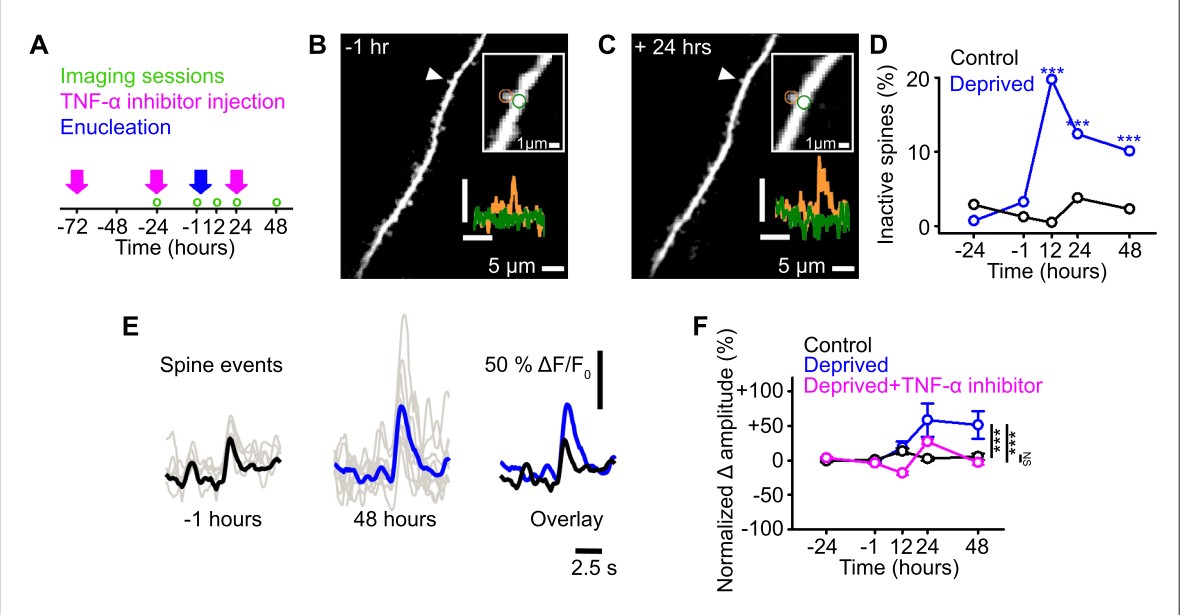

**Figure 1.** Functional strengthening of dendritic spines following sensory deprivation. (**A**) Experimental timeline. Green circles denote imaging time points, pink arrows denote time of tumor necrosis factor alpha (TNF-α) inhibitor injections, and blue arrow shows time of monocular enucleation (0 hr). (**B,C**) Maximum intensity projection of activity for a dendritic stretch of a layer 5 pyramidal cell measured before (**B**, –1 hr) and after (**C**, 24 hr) deprivation. Upper inset, higher magnification image of highlighted (arrowhead) individual spine, in which circles show the region of interest for the dendritic spine (orange) and the dendrite (green). Lower inset, calcium fluorescence signals measured at the dendritic spine (orange) and adjacent dendrite (green). Axons were removed from the background of the image for clarity. Scale bar, 25% $\Delta F/F_0$, 5 s. (**D**) Percentage of spines that become inactive at each time point in control (black) and deprived (blue) mice. All spines included were active during at least one imaging time point. Asterisks denote comparison of deprived and control values. (**E**) Single trial (gray) and average (black/blue) events, aligned by peak, of an example spine at –1 hr (left, black) and 48 hr (middle, blue) from deprivation (0 hr), and the overlay of both (right). (**F**) Percent change in amplitude normalized to baseline (–24 and –1 hr) of calcium transients measured in persistent (active at all time points) dendritic spines in control (black), deprived (blue) and deprived with TNF-α inhibitor (pink) conditions. Deprivation occurs at 0 hr. For all statistics, see ***Figure 1—source data 1***. ***p<0.001, NS, no significance. Error bars, standard error of the mean (SEM).

The online version of this article includes the following source data and figure supplement(s) for figure 1:

**Source data 1.** Statistical comparisons for *Figure 1*.

**Figure supplement 1.** Spine event frequency.

**Figure supplement 1—source data 1.** Statistical comparisons for *Figure 1—figure supplement 1*.

## Functional strengthening of dendritic spines following sensory deprivation

Consistent with previous work (***Barnes et al., 2017***), we found that approximately 20% of spines in the contralateral visual cortex became inactive 12 hr after monocular deprivation, with the number of inactive spines decreasing over time (***Figure 1D***, see Methods). In these experiments, inactive spines could represent either inactivity or spine loss, which can be indistinguishable with GCaMP6s labelling. On average, the persistent (active at all time points) spines increased their functional response amplitudes (***Figure 1E–F***). It is important to note that the reduction of inactive spines over time may reflect spines whose activity decreased below detection thresholds immediately following deprivation and then, following subsequent homeostatic strengthening, were detectable again.

Certain forms of homeostatic synaptic changes are dependent on the cytokine tumor necrosis factor alpha (TNF-α) (***Beattie et al., 2002***; ***Stellwagen and Malenka, 2006***), so we examined if that was the case for the synaptic strengthening observed here. The increased amplitude of spine responses was blocked by an intraperitoneal injection of a dominant negative form of TNF-α (***Figure 1A and F***), suggesting this effect is dependent on TNF-α. Following deprivation, we found no changes over time in the average frequency of responses at the population level (***Figure 1—figure supplement 1E***).

Previous work has shown that spine-specific decreases in response frequency drive synapse-specific increases in response amplitude (***Lee et al., 2010***). We therefore examined if frequency changes were

indicative of amplitude changes in individual spines. We observed no correlation between changes in frequency and amplitude within individual spines (*Figure 1—figure supplement 1F*), indicating that the changes in frequency are not likely to be driving the changes in amplitude within a single spine.

## Identifying functional subsets of dendritic spines in visual cortex

We have previously used structural imaging of spine size as an in vivo proxy for synaptic strength and shown that not all spines undergo a TNF-α-dependent strengthening following deprivation (*Barnes et al., 2017*). One possibility is that there could be common functional properties of spines that undergo homeostatic strengthening. We therefore characterized the response properties of individual spines prior to any form of deprivation during the baseline time points (–24 and –1 hr). We first measured the responses of spines to visual stimuli: bilateral drifting gratings (*Barnes et al., 2015*) and a sparse noise stimulus (*de Vries et al., 2020*; *Figure 2A*). Spines that had time-locked responses to the visual stimulus at levels above chance (see Methods) (*Barnes et al., 2015*; *Ko et al., 2011*) were categorized as 'visually responsive' spines (*Figure 2B*). We found that a majority of spines that were responsive to the sparse noise stimulus also responded to drifting gratings (97%). Therefore, in a subset of experiments, we characterized visually responsive spines using only the drifting gratings stimulus to allow sufficient imaging time to test other stimuli. When we examined the functional strengthening of all of the spines, we found an inverse relationship, such that the spines with the lowest visual responsiveness prior to deprivation underwent the strongest functional strengthening after deprivation (*Figure 2C*). Consistent with our previous work (*Barnes et al., 2017*), we observed that only a subset of spines increased their functional responses (*Figure 2C*).

We next characterized other functional inputs. Within mouse visual cortex, there are high levels of correlated network activity, which may reflect the intrinsic dynamics (*Okun et al., 2015*; *Stringer et al., 2019*). We therefore hypothesized that activity in some spines may reflect this network activity, potentially via local intracortical connections (*Okun et al., 2015*). We measured the correlation of individual spines' activity with the 'network signal', which we measured as the average activity during all experimental conditions (visual stimulation and darkness) of all the other spines in the same imaging region (*Okun et al., 2015*; *Figure 2A*, see Methods). The network signal does not reflect locomotion-related activity, as the mice were stationary during our measurements.

In our population of spines, we found that most spines (89%) had a significant correlation with the network signal. This included the visually responsive spines, all of which had a significant correlation with the network signal (see Methods). As described above, all spines with time-locked responses to visual stimuli at above chance levels (see Methods) were classified as visually responsive spines (24%; *Figure 2B*), even though their activity also correlated with the network signal. Spines with activity that correlated with the network signal, but did not have visually-evoked responses, were classified as network-correlated (65%). The remainder of spines neither responded to the visual stimuli nor correlated with the network activity, which we refer to as unclassified (11%; *Figure 2A*). We tested whether these unclassified spines were correlated with other behavioral measures, such as pupil movement, pupil dilation (diameter), and whisker movement, and found no evidence of significant correlations above chance (see Methods). There were no significant differences in the average activity in the three groups of spines prior to deprivation (measured as the area under the curve of the $\Delta F/F_0$ fluorescence signal; comparison between network-correlated, visually responsive and unclassified spine activity, one-way ANOVA, p=0.232).

We next examined if input types were clustered on the same dendritic branches (*Hedrick et al., 2022*). All of the branches that we measured had all three types of spines (visually responsive, network-correlated, unclassified). Within individual dendritic branches, we found that the local spatial organization of functional response properties (visually responsive, network-correlated, unclassified) of the dendritic spines were not different from chance (*Figure 2D–F*, see Methods).

## Functional strengthening occurs in network-correlated spines

We next examined if the spines that were functionally strengthened had common functional properties (*Figure 3*). Consistent with our earlier finding (*Figure 2C*), we found no evidence of an increase in the amplitude of the persistent spines identified as visually responsive (*Figure 3A*), or of the persistent unclassified spines (*Figure 3B*). However, following deprivation there was an increase in the response amplitude of persistent network-correlated spines (*Figure 3C*). Overall activity depends on

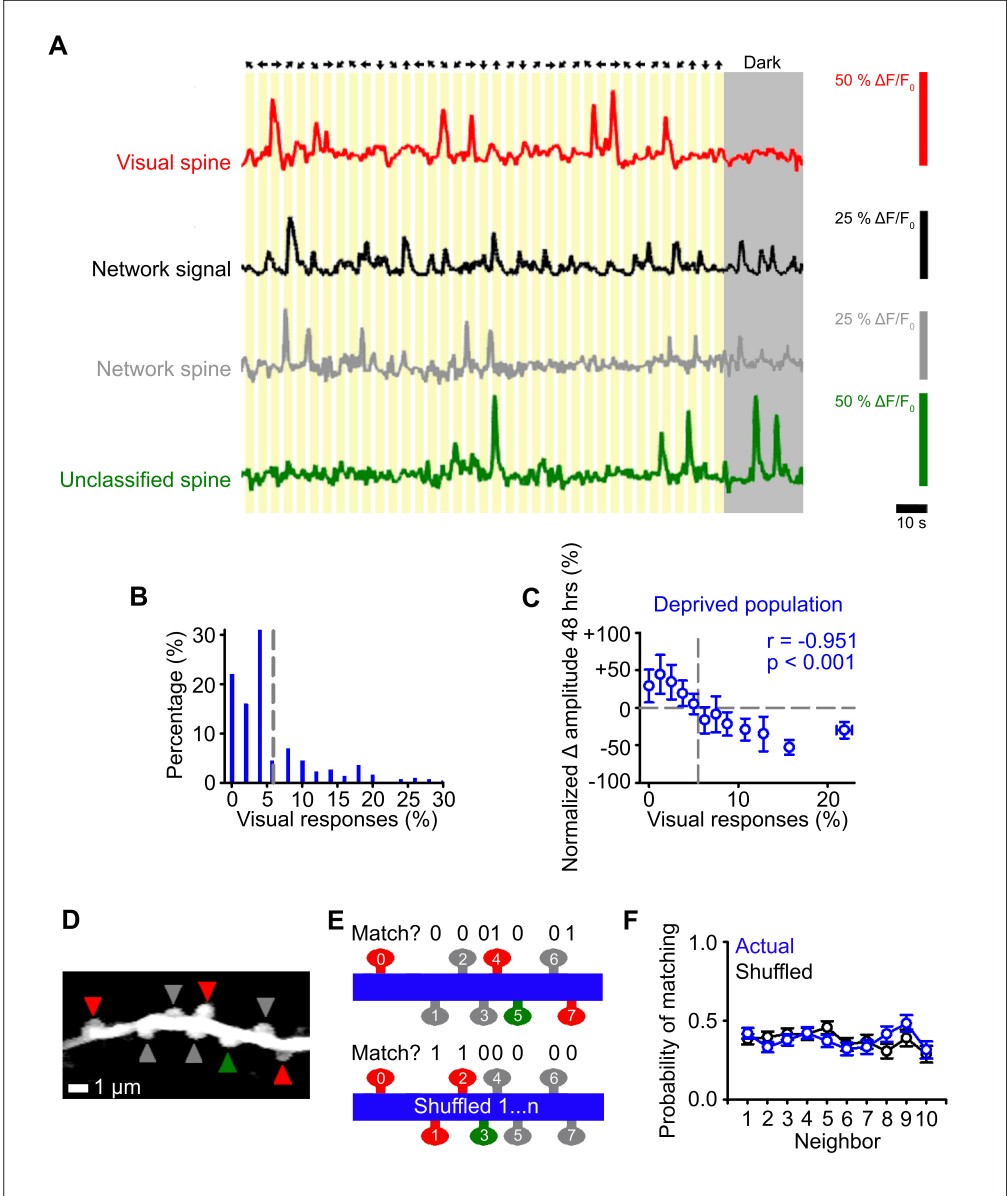

**Figure 2.** Identifying functional subsets of dendritic spines in visual cortex. (**A**) Schematic of the stimulus orientation and direction (top) and calcium responses of an individual visually responsive spine (red), the average network response (black), an individual network-correlated spine (gray), and an individual unclassified spine (green). Dark gray shading indicates the mouse was stationary in the dark. (**B**) Distribution of the percentage of visual stimuli that elicit a time-locked response for all deprived spines during baseline (–24 and –1 hr). Gray dashed line indicates the number of time-locked responses to visual stimuli expected by chance (5.5%, see Methods). (**C**) Percent change in spine amplitude at 48 hr normalized to baseline (–24 and –1 hr) for the entire population of spines in deprived mice as a function of the percent of visual stimuli out of all visual presentations that elicit time-locked responses, measured during baseline (–24 and –1 hr). Gray dashed line indicates chance levels (see Methods) for percentage of visual responses (vertical) and no change in spine amplitude relative to baseline (horizontal). (**D**) Maximum intensity projection of activity of a dendritic segment with network-correlated (gray), visually responsive (red), and unclassified (green) spines. (**E**) Schematic of functional property clustering analysis for first iteration of $spine_0$ (top) and shuffled condition (bottom), for the example dendrite in Panel D. Spines match (value of 1) when they have the same functional class (network-correlated [gray], visually responsive [red], unclassified [green]) as $spine_0$. (**F**) Population data for probability of matching the functional properties of a neighboring spine for the actual data (blue) and randomly shuffled position (black) of a given spine on a dendritic branch. For statistics, see *Figure 2—source data 1*. Error bars, standard error of the mean (SEM).

The online version of this article includes the following source data for figure 2:

*Figure 2 continued*

**Source data 1.** Statistical comparisons for *Figure 2*.

both the amplitude and frequency of responses. While the average frequency of responses did not change following deprivation (*Figure 1—figure supplement 1E*), we examined if functional groups of spines had changes in response frequency and observed a decrease in the frequency of responses of visually responsive spines over time following deprivation (*Figure 3—figure supplement 1A–C*). When we examined the integral of the activity trace (which combines amplitude and frequency) over time following deprivation, we observed a decrease for visually responsive spines (*Figure 3—figure supplement 1D*), no change for unclassified spines (*Figure 3—figure supplement 1E*), and an increase for network-correlated spines (*Figure 3—figure supplement 1F*). We also found a strong

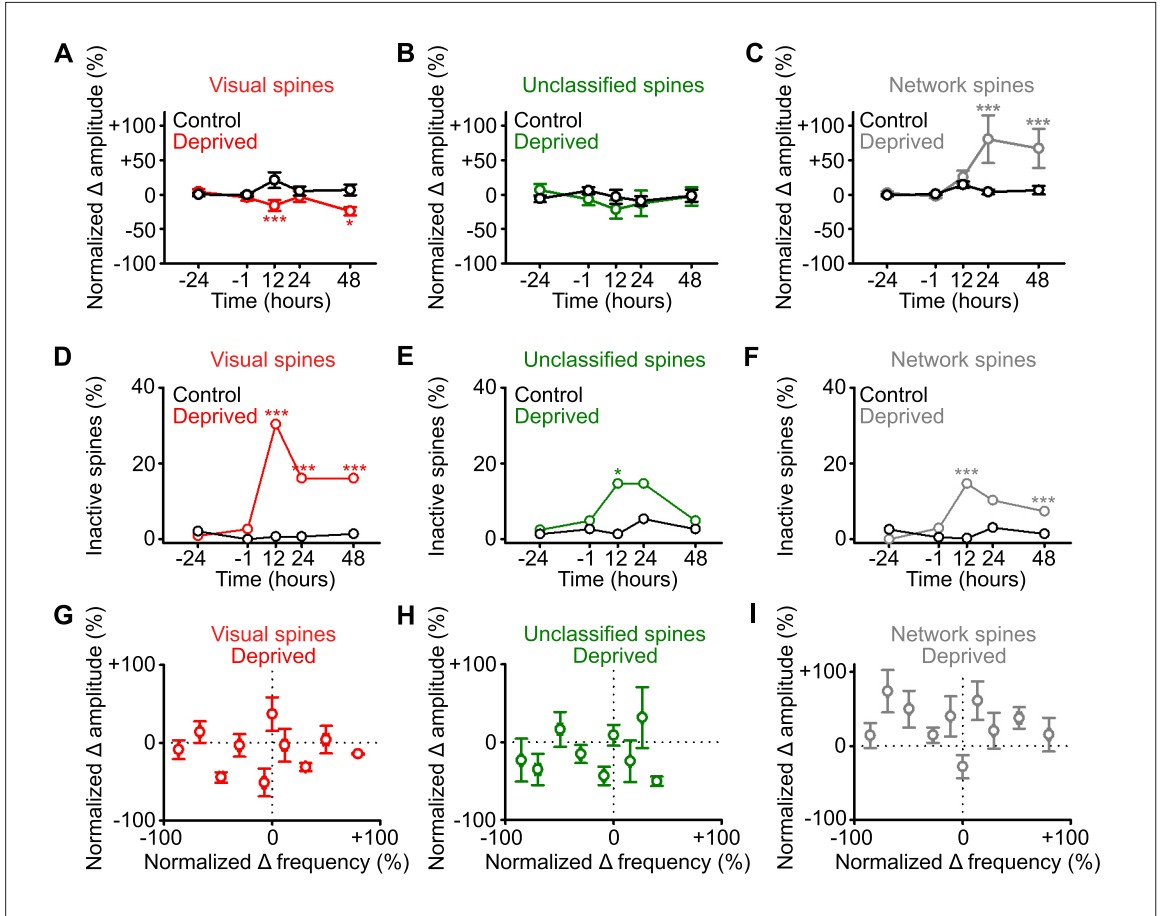

**Figure 3.** Functional strengthening occurs in network-correlated spines. (**A,B,C**) Percent change in spine amplitude normalized to baseline (–24 and –1 hr) of calcium transients for persistent (active at all time points) spines that were classified during baseline as visually responsive (**A**, red), unclassified (**B**, green), and network-correlated (**C**, gray), either following enucleation (0 hr) or sham-enucleated controls (black). (**D,E,F**) Percentage of spines that become inactive that were classified during baseline as visually responsive (**D**, red), unclassified (**E**, green), and network-correlated (**F**, gray), either following enucleation (0 hr) or sham-enucleated controls (black). All spines included are active during at least one imaging time point. (**G,H,I**) In deprived mice, normalized percent change in calcium transient frequency at 12 hr (relative to baseline: –24 and –1 hr) versus normalized percent change in amplitude at 48 hr (relative to baseline: –24 and –1 hr) for individual persistent (active at all time points) spines classified as visually responsive (**G**), unclassified (**H**), and network-correlated (**I**). For statistics, see *Figure 3—source data 1*. *p<0.05, ***p<0.001. Error bars, standard error of the mean (SEM).

The online version of this article includes the following source data and figure supplement(s) for figure 3:

**Source data 1.** Statistical comparisons for *Figure 3*.

**Figure supplement 1.** Network-correlated spines are functionally strengthened in visual cortex.

**Figure supplement 1—source data 1.** Statistical comparisons for *Figure 3—figure supplement 1*.

increase in the fraction of inactive visually responsive spines after deprivation. While a larger fraction of visually responsive spines become inactive, the absolute number of network-correlated spines that became inactive was higher, since network-correlated spines reflect a larger proportion of the total spine population. These results suggest that the increase in spine loss previously reported following enucleation (*Barnes et al., 2017*) may include both spines that were visually responsive and network-correlated prior to deprivation (*Figure 3D–F*). A sizeable fraction of visually responsive spines remains active after visual deprivation (*Figure 3D*). It is important to note that following sensory deprivation, we no longer observed responses to visual stimuli in the visually responsive spines, since the feed-forward input had been permanently removed. The responses of the previously categorized visually responsive spines consisted of network responses following deprivation.

There was no correlation between amplitude and frequency within individual spines in any of the groups (*Figure 3G–I*), again suggesting that frequency decrease does not regulate changes in amplitude within individual spines in this paradigm. Overall, these results suggest the network-correlated inputs, and not the identified visually responsive inputs, are homeostatically strengthened following sensory input loss.

## Identified sensory responsive spines are not functionally strengthened

Having found that the network-correlated spines increase their activity, we wanted to determine whether this more generally reflected the functional strengthening of all non-deprived inputs. To investigate this issue, we used a multimodal brain area – the retrosplenial cortex (RSC) – which receives measurable auditory responsive and visually responsive inputs (*Figure 4A*). We again measured responses of dendritic spines on layer 5 cells in the distal tufts in layer 2/3. Using the same classification approach as before (*Figure 2*, see Methods), we found that the activity of 92% of the spines was correlated with the network signal. Within this group, a subset of spines had time-locked responses to visual (18.7%) or auditory (15.9%) stimuli (*Figure 4A*) and the rest correlated only with the network signal (57.4%, *Figure 4A*). The remaining 8% of spines were unclassified (no correlation with the network signal and no sensory-evoked responses). Following reversible visual deprivation via dark exposure for 48 hr (*Figure 4B*), we observed an increase in the average spine response amplitude across the population. This response was TNF-α-dependent (*Figure 4C*), similar to what we observed in visual cortex following enucleation (*Figure 1*). When we examined the changes in the responses of functionally identified spines, we observed an increase in the response amplitude of the network-correlated spines, but not in the visually responsive or unclassified spines (*Figure 4D*, *Figure 4—figure supplement 1A–C*), consistent with our results in visual cortex (*Figure 2C*, *Figure 3A–C*). We then examined the responses of the auditory responsive spines, which represent an identified non-deprived sensory input. After visual deprivation, the auditory responsive spines did not undergo functional strengthening but maintained their baseline amplitude following visual deprivation (*Figure 4E*).

To ensure that the strengthening effects that we observed were not exclusive to visual deprivation, we performed a complementary, but separate, experiment where we deprived mice of auditory input through reversible ear plugging for 48 hr and again measured post-deprivation spine responses in the RSC (*Figure 4F*). We observed a TNF-α-dependent population increase in spine responses (*Figure 4G*). Similar to visual deprivation, network-correlated spines increased their responses following auditory deprivation (*Figure 4H–I*, *Figure 4—figure supplement 1D–F*), but there was no change in the response strength of either (non-deprived) visually responsive (*Figure 4H*) or (deprived) auditory responsive spines (*Figure 4I*). Together, these data suggest that increases in spine response amplitude predominantly occur in the spines whose activity correlated with the network signal and not in the spines which are responsive to the non-deprived or deprived sensory stimulation.

## Increased global sensory-evoked responses correlate with increased network-correlated spine responses

Previous work has shown that following visual deprivation, there is a homeostatic increase in the cellular visually-evoked responses relative to baseline (*Mrsic-Flogel et al., 2007*). Critically, this homeostatic response is hypothesized to occur through a presumed strengthening of the spared visual inputs (*Kaneko et al., 2008*; *Keck et al., 2017*; *Mrsic-Flogel et al., 2007*; *Pandey et al., 2022*; *Turrigiano, 2017*). Given that we did not observe increases in the response strength of sensory responsive spines, we examined if the global sensory responses were increased following dark exposure (a reversible

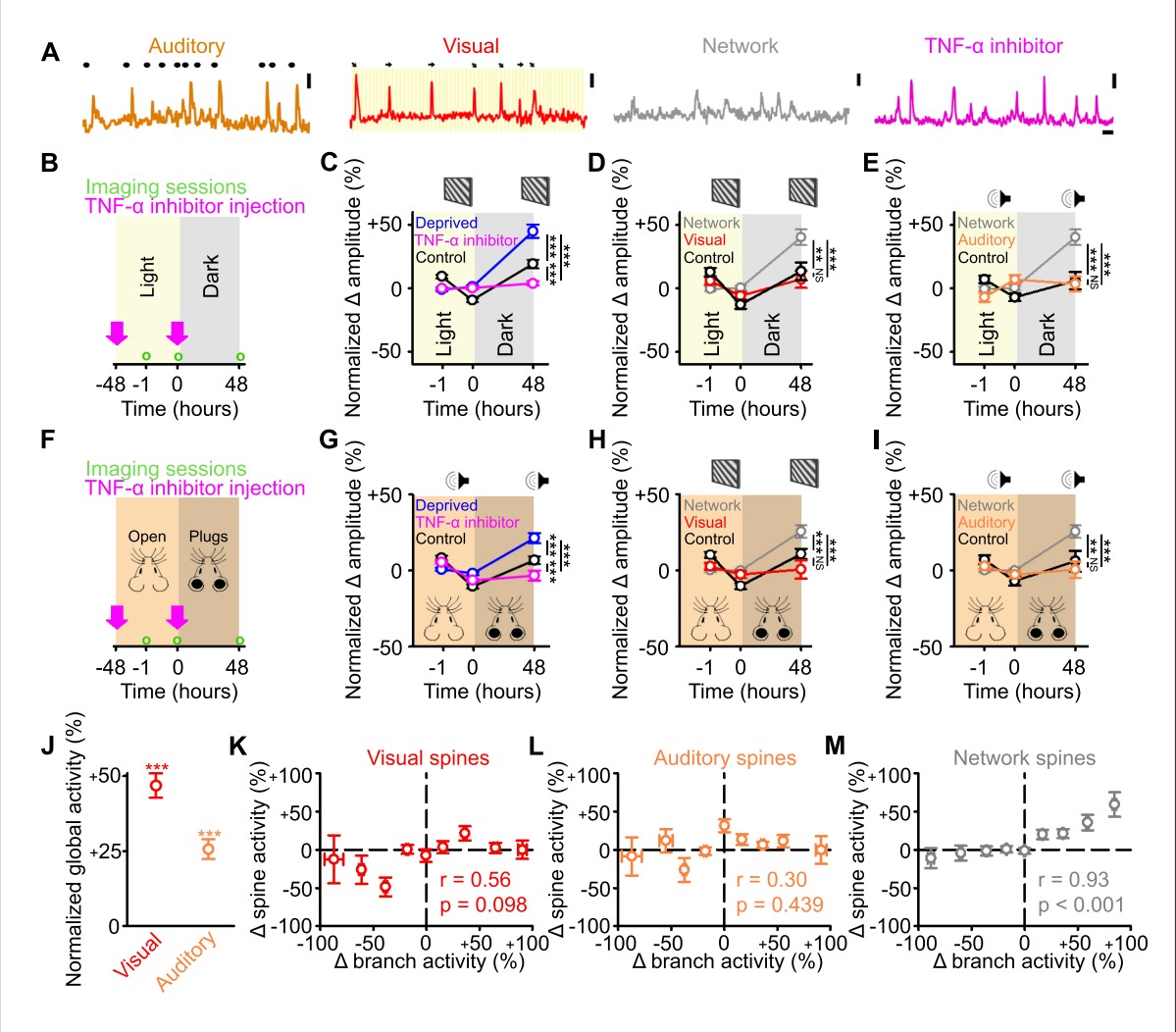

**Figure 4.** Identified sensory responsive spines are not functionally strengthened after sensory deprivation. (**A**) Example activity of auditory responsive (orange), visually responsive (red), and network-correlated (gray) spines, as well as following tumor necrosis factor alpha (TNF-α) inhibitor injections (pink). Auditory: black dots above the trace indicate the timing of auditory stimuli. Visual: orientation and direction of presented visual stimuli that evoked a response are shown above the trace. Scale bars, 50% ΔF/F$_0$ and 10 s. (**B,F**) Experimental timeline for dark exposure (**B**) or auditory ear plug (**F**) deprivation experiments showing imaging time points (green circles) and timing of TNF-α inhibitor injections (pink arrows). (**C,G**) Percent change in amplitude normalized to baseline (–1 and 0 hr) of all spines following visual (**C**) or auditory (**G**) deprivation (blue), control (black) or deprivation with TNF-α inhibition (pink) in response to drifting gratings (**C**) or auditory stimulation (**G**). (**D,H**) During presentation of drifting gratings, percent change in amplitude normalized to baseline (–1 and 0 hr) for control visually responsive spines (black), deprived network-correlated spines (gray), or deprived visually responsive spines (red) following visual deprivation (**D**) or auditory deprivation (**H**). (**E,I**) During auditory stimulation, percent change in amplitude normalized to baseline (–1 and 0 hr) for control auditory responsive spines (black), deprived network-correlated spines (gray), or deprived auditory responsive spines (orange) following visual deprivation (**E**) or auditory deprivation (**I**). (**J**) Forty-eight hours after visual deprivation, average global response amplitude (measured in dendritic branches) in response to drifting gratings (red) or auditory stimulation (orange), normalized to the response of the same branches to the same stimulus during baseline (–1 and 0 hr). (**K,L,M**) Change in activity 48 hr after visual deprivation normalized to baseline (–1 and 0 hr) for parent global dendritic branch activity and for spine activity for visually responsive (**K**), auditory responsive (**L**), and network-correlated (**M**) spines on that branch. For statistics, see *Figure 4—source data 1*. **p<0.01, ***p<0.001, NS, no significance. Error bars, standard error of the mean (SEM). Asterisks in C–I denote comparisons at 48 hr.

The online version of this article includes the following source data and figure supplement(s) for figure 4:

**Source data 1.** Statistical comparisons for *Figure 4*.

**Figure supplement 1.** Network-correlated spines are functionally strengthened in retrosplenial cortex (RSC).

**Figure supplement 1—source data 1.** Statistical comparisons for *Figure 4—figure supplement 1*.

deprivation paradigm) in adult mice. As a proxy for spiking, we measured the global dendritic signals from the same dendrites of layer 5 cells located in lower layer 2/3, where we also measured the spine signals, which we refer to as the 'global dendritic responses'. In sensory cortices under our experimental conditions, the majority of dendritic signals are thought to reflect back-propagating action potentials (*Beaulieu-Laroche et al., 2019*; *Francioni et al., 2019*). We found that following visual deprivation via dark exposure, the global dendritic responses in RSC to both visual and auditory stimulation were increased when compared with baseline measurements (*Figure 4J*), consistent with past results (*Mrsic-Flogel et al., 2007*). This was despite the fact that neither the visually responsive nor auditory responsive spines increased their responses to the same respective stimuli (*Figure 4D–E*). Both the global visually-evoked responses and individual visually responsive spines maintained their visual tuning before and after deprivation at levels comparable to control (*Figure 4—figure supplement 1—source data 1*).

We then examined the relationship between changes in post-deprivation activity relative to pre-deprivation baseline activity for both global dendritic responses and visually-evoked spine responses. Increases in global dendritic responses after deprivation were uncorrelated with the changes in visually-evoked spine responses on those dendrites, making it unlikely that increases in global dendritic responses were driven by increases in the strength of visually-evoked spine responses (*Figure 4K*). There was also no relationship within a dendrite between changes in global dendritic responses and auditory-evoked spine responses (*Figure 4L*). Conversely, increases in global dendritic responses were correlated with an increase in the network-correlated spine activity on the same dendrites (*Figure 4M*). Our results suggest that increases in sensory-evoked global dendritic responses are unlikely to result from increases in the response strength of the identified sensory responsive inputs.

## Discussion

Here, we report a TNF-α-dependent increase in the functional responses of dendritic spines in awake mice following sensory deprivation. Surprisingly, when we examine the functional properties of spines that undergo strengthening, we find dendritic spines whose activity is correlated with the network signal are strengthened, but spines identified as sensory responsive are not. This strengthening of network-correlated spines occurs over a similar time course and magnitude in both visual cortex and RSC and in response to different deprivation paradigms. Despite the fact that we do not see an increase in responses of sensory responsive spines, we do observe a homeostatic increase in global sensory-evoked responses following sensory deprivation. This global increase is correlated with an increase in the response strength of network-correlated spines on those branches, but not of identified sensory responsive spines. Our findings suggest that in vivo synaptic homeostasis during adulthood occurs in a functional subset of inputs, specifically those correlated only with network activity, rather than the identified sensory responsive inputs that are thought to drive sensory processing.

### Homeostatic regulation of network activity

Previous work has hypothesized that inputs are homeostatically strengthened following sensory deprivation (*Keck et al., 2017*; *Turrigiano, 2017*), and that this process may facilitate the homeostatic regulation of cellular activity levels (*Barnes et al., 2015*; *Hengen et al., 2016*; *Hengen et al., 2013*; *Keck et al., 2013*). Our results suggest that the homeostatic increase in sensory-evoked responses observed after deprivation in adult mice is not associated with increases in the response strength of identified sensory responsive inputs. Instead, we observe a correlation between increases in global sensory-evoked activity levels and increases in the functional response strength of network-correlated spines. Why do we see an increase in global sensory-evoked responses if there is not a change in the functional strength of the sensory responsive inputs? One possibility is that the strengthened network connections serve to amplify intrinsic activity, keeping cells closer to the threshold for action potential generation and thereby making incoming sensory responsive inputs more likely to elicit the observed sensory-evoked increases in post-synaptic activity. In line with this idea, previous work has shown that homeostatic recovery at the level of small networks of cells is mediated by increases in the intrinsic network activity (*Barnes et al., 2015*) and that intrinsic network activity is amplified within these interconnected neural ensembles (*Miller et al., 2014*). Together, these findings suggest that the amplification of network activity through a strengthening of network connections may underlie rebalancing of

activity in the cortex following sensory deprivation affecting network activity. This provides a mechanism during adulthood by which overall activity can be modulated through the network inputs, without altering the identified sensory inputs and the established circuits thought to drive sensory processing.

If intrinsic network activity is amplified, what might these network-correlated inputs encode? Network-correlated spines do not respond: (1) in the case of visual stimulation to either drifting gratings or sparse noise visual stimuli, which are thought to drive broad visual responses, or (2) in RSC, in the case of auditory stimulation, specifically white-noise bursts. We cannot rule out that these network-correlated spines may respond to more specific visual or auditory stimuli that were not presented, or to sensory-feedback signals from higher brain regions, which may not be time-locked to the visual stimuli. The presence of the unclassified inputs as a category indicates that (1) not all inputs correlate with the network signal by default and (2) network-correlated inputs do not simply encompass inputs that were not identified as sensory responsive. The mouse cortex is known to respond to many non-sensory inputs and the network signal could reflect contextual signals (*Khan and Hofer, 2018*; *Pakan et al., 2016*; *Pakan et al., 2018b*; *Tohmi et al., 2014*; *Yoshitake et al., 2013*), spatial information (*Fiser et al., 2016*; *Pakan et al., 2018a*; *Saleem et al., 2018*), intrinsic signals (*Miller et al., 2014*; *Okun et al., 2015*; *Stringer et al., 2019*), or intracortical inputs, which are known to be preferentially strengthened in ex vivo preparations following dark exposure (*Petrus et al., 2015*). The network signal likely represents some combination of these signals. Given that our data were collected while the mice were stationary, it is unlikely that the network signal is driven by activity related to locomotion.

## Synaptic and molecular mechanisms

We find evidence that the functional strengthening of network-correlated spines is dependent on the cell-signalling cytokine TNF-α. Previous work has shown that soluble TNF-α is released from glial cells following reductions in glutamate levels and plays a key role in synaptic strengthening by facilitating the insertion of α-amino-3-hydroxy-5-methyl-4-isoxazolepropionic acid receptors (AMPAR) (*Stellwagen and Malenka, 2006*). Importantly, the synaptic localization of *N*-methyl-D-aspartate receptors (NMDAR) and their associated currents are unaffected by changes in TNF-α levels (*Beattie et al., 2002*; *He et al., 2012*). The specific actions of TNF-α may in part explain in vivo findings, where pharmacological blockade of TNF-α has shown selective impairments of homeostatic, but not NMDAR-dependent, Hebbian-like plasticity processes (*Kaneko et al., 2008*). Further work is required to better understand the key neural circuit and molecular features that allow the selective modulation of homeostatic adaption and even more specifically, the network-correlated spines. One possibility is that TNF-α could target specific receptor profiles, potentially through differential synaptic expression of the TNF receptor 1 which is thought to be central to TNF-α-dependent synaptic plasticity (*Grell et al., 1995*; *He et al., 2012*); however, this would assume that homeostatic regulation occurs in the layer 5 cells.

Previous work has shown that TNF-α-dependent AMPAR insertion during homeostatic regulation is associated with the glutamate receptor-interacting protein 1 (GRIP1), a synaptic scaffolding protein which facilitates insertion of synaptic AMPARs, thereby increasing the AMPAR current and number (*Barnes et al., 2017*; *Gainey et al., 2009*). Increases in GRIP1-dependent AMPAR insertion may be involved in the functional strengthening of spine responses that we observe here, as we have previously demonstrated that AMPAR-mediated synaptic strengthening occurs following monocular enucleation in a TNF-α-dependent manner and involves GRIP1 (*Barnes et al., 2017*). Alternatively, changes in inhibitory neural circuitry may contribute to the increased functional response strength that we observe in dendritic spines. Previous work has demonstrated that inhibitory synapses located on dendritic spines (also containing excitatory synapses) are eliminated at a higher rate following sensory deprivation (*Chen et al., 2012*; *van Versendaal et al., 2012*). The loss of local synapse-specific inhibition could enhance the response strength of individual spines; however, given that these changes in inhibitory synapses are only seen in a subset of spines (~15%), they are unlikely to be solely responsible for the observed amplitude increases in the network-correlated spines we report, which comprise more than 60% of the total spine population.

Finally, it is important to note that while we made these measurements in layer 5 pyramidal cells, the homeostatic changes mediated by TNF-α could occur outside of layer 5, including changes to upstream inputs or changes to the presynaptic responses, either through changes in presynaptic release (*Vitureira et al., 2012*) or through a change in activity patterns of the presynaptic cell (e.g.,

bursts compared to single spikes) (*Linden et al., 2009*). Given that both sensory responsive spines (which do not strengthen) and network-correlated spines (which do strengthen) exhibit only network-correlated responses after sensory deprivation, changes in presynaptic activity patterns of network signals would require targeted changes to the network-corelated spines, but not the previously sensory responsive spines. Further research is necessary to identify the role of presynaptic plasticity in homeostatic synaptic plasticity in vivo.

## Network homeostasis and sensory processing

If the network-correlated inputs are homeostatically regulated, what happens to the sensory responsive inputs following deprivation? We found evidence for a reduction in the overall activity of identified sensory responsive spines in visual cortex, with a fraction of visually responsive spines becoming inactive following enucleation. Our previous work has shown that monocular enucleation is associated with structural spine loss, suggesting that part of the functional silencing we observe here may occur through loss of dendritic spines (*Barnes et al., 2017*). A considerable number of identified sensory responsive spines do remain active, but in both visual cortex and RSC, they undergo no increase in response amplitude following sensory deprivation, independent of whether they are deprived or non-deprived inputs. Thus, we consistently observe that identified sensory responsive inputs are not strengthened in sensory cortices following deprivation, but that many sensory responsive inputs are preserved, which could underlie the stability of visual tuning that has been observed following sensory deprivation (*Figure 4—figure supplement 1—source data 1*; *Rose et al., 2016*).

What are the potential advantages of strengthening the network-correlated inputs, rather than the sensory responsive inputs? Through homeostatic regulation of the network activity, feedforward sensory input-output relationships that may encode sensory information and underlie sensory processing are not perturbed during adulthood. Our previous work (*Barnes et al., 2017*) has suggested that scaling up the strongest sensory inputs could lead to a saturation of these inputs, due to known dendritic non-linearities in the local integration of multiple inputs (*Branco and Häusser, 2011*; *Branco and Häusser, 2010*). If sensory responsive inputs are saturated and no longer have distinct weights, information processing in the neuron is reduced (*Barnes et al., 2017*). By homeostatically regulating the network-correlated inputs, the dynamic range of the sensory responsive inputs – both the remaining deprived sensory responsive inputs and the non-deprived sensory responsive inputs – is unaltered and the cortex can continue to process a wide range of external sensory stimuli (*Panzeri et al., 2017*; *Panzeri et al., 2010*).

# Methods

**Key resources table**

| Reagent type (species) or resource | Designation | Source or reference | Identifiers | Additional information |
| --- | --- | --- | --- | --- |
| Strain, strain background (*AAV*) | AAV2/1-ef1a-GCaMP6s | FMI vector core | N/A | |
| Strain, strain background (*Mus musculus*) | C57BL/6J | Charles River | N/A | |
| Chemical compound, drug | XPRo1595 | Xencor, Inc | N/A | |

## Mice

Experiments were conducted according to the United Kingdom Animals (Scientific Procedures) Act 1986 or were approved by the Veterinary Department of the Canton of Basel-Stadt, Switzerland. We used 43 adult (P60-120) male and female mice (C57BL/6) and measured 5858 individual spines. All mice were sex and age matched within experimental groups. Mice were housed with littermates (two to six mice depending on litter size) and kept on a 12 hr light-dark cycle at 21°C. Imaging experiments were time matched during the light cycle (except in dark exposure experiments).

## Surgery

For in vivo imaging experiments, cranial windows were surgically implanted, as described previously (*Holtmaat et al., 2009*), over the right hemisphere of visual cortex (coordinates 1.5–2.5 mm lateral of the midline and 0–1 mm anterior of lambda), which corresponds to the monocular region of visual cortex, or over the right hemisphere of RSC (coordinates 0.2–0.8 mm lateral of the midline and 0–1.5 mm anterior of lambda). We made a craniotomy in ketamine/xylazine (0.15 and 0.015 mg/g of body weight, respectively) anesthetized mice and replaced the skull with a glass coverslip that was attached to the bone with dental cement. Mice were injected with AAV2/1-ef1a-GCaMP6s into layer 5 (500 µm deep) before the glass coverslip was positioned. Mice were allowed to recover for at least 28 days after surgery before imaging commenced. For monocular enucleation, we applied lidocaine (Emla cream) to the area around the left (contralateral) eye in anesthetized mice prior to surgical removal of the eye. Control sham-enucleated mice were given time-matched anesthesia. For dark exposure experiments, mice were housed in a dark cabinet for 48 hr. For auditory deprivation experiments, mice were anesthetized with isoflurane and the ear canal was filled with a plastic polymer (Kwik-Sil) (*Zhuang et al., 2017*) for 48 hr. The polymer was removed prior to testing at the 48 hr time point. Control sham-deprived mice were given time-matched anesthesia. Mice were randomly assigned into deprivation or sham-deprivation groups, such that half of each litter was in each group (except for dark exposure, where assignments were randomly made by cage, so as to not separate littermates). In visual cortex experiments, mice were imaged at –24 and –1 hr prior to deprivation or sham-deprivation and 12, 24, and 48 hr after. For RSC experiments, mice were imaged twice in a baseline session and then 48 hr later following deprivation or sham-deprivation. For experiments with the TNF-α inhibitor, mice were injected with XPro1595 (*Barnes et al., 2017*; *Lewitus et al., 2014*) (Xencor, Inc) at a dose of 10 mg/kg at –48 and –1 hr for RSC experiments and at –72, –24, and 24 hr for visual cortex experiments.

## Functional imaging and sensory stimulation

Measurements of functional imaging data were conducted as described previously (*Barnes et al., 2017*; *Barnes et al., 2015*; *Sammons et al., 2018*). Functional calcium imaging of volumes of visual cortex or RSC was performed on a custom-built two-photon microscope with a 12 kHz resonance scanner (Cambridge Technology) and a high-power objective Z-piezo stage (Physik Instrumente), using a MaiTai eHP laser with a DeepSee prechirp unit (Newport/Spectra Physics) set to 910 nm and a Nikon 16×, 0.8 NA objective, as described previously (*Barnes et al., 2015*). Data were acquired with an 800 MHz digitizer (National Instruments) and pre-processed with a custom programmed field programmable gate array (National Instruments). The dynamic range of both the amplifier and the photo-multiplier tube exceeded the digitization range and the data acquisition software automatically detected digital saturation of all pixels. Imaging data were full-frame registered using a custom written registration algorithm. To remove slow signal changes in raw fluorescence traces, the 8th-percentile value of the fluorescence distribution in a ±15 s window was subtracted from the raw fluorescence signal (*Dombeck et al., 2007*). Intrinsic signal imaging before enucleation was used to localize the monocular visual cortex as described previously (*Barnes et al., 2015*; *Keck et al., 2013*), and the RSC was localized using stereotaxic coordinates. Layer 5 dendrites were identified by tracing the dendrites to the cell bodies. Separating layer 2/3 from layer 5 was aided by the fact that expression levels in layer 4 neurons were much lower in our preparation.

Mice were allowed to habituate to the setup while head-fixed and data were included from periods when the mice were stationary. Mice were presented with drifting gratings in eight directions (0–360 degrees in 45 degree steps) presented in a random order, with a spatial frequency of 0.04 cycles/degree and a temporal frequency of 2 Hz. In a subset of mice, we also presented a sparse visual noise stimulus. This consisted of a sequence of sparse noise stimuli that changed every 250 ms. Each sparse noise stimulus was a set of between 8 and 12, black or white squares (of 8 degrees of visual angle) on a gray background. The location of the squares was chosen randomly, but such that squares did not overlap in an individual sparse noise stimulus (*de Vries et al., 2020*). Auditory stimulation consisted of the presentation of bilateral white noise bursts at 65 dB presented 30 cm from the mouse's ears, with timing according to a uniform distribution (minimum interval of 3 s and maximum interval of 5 s). Prior to experiments, we chose a noise volume that was audible to the mice (*Meyer et al., 2018*). Mice were also imaged for a 6 min period in the dark, without auditory stimulation. The

same stimulus paradigms were used at each time point before and after enucleation, dark exposure, auditory deprivation, or sham-deprivation.

## Spine signal extraction

Calcium responses from individual spines were isolated from global dendritic signals using a subtraction procedure described previously (*Chen et al., 2013b*; *Iacaruso et al., 2017*; *Wilson et al., 2016*). To measure spine signals, circular ROIs were first drawn over individual dendritic spines to measure spine fluorescence and compute $\Delta F/F_{0\_spine}$. We then drew circular ROIs of the same size as the spine ROI around the adjacent parent dendrite to the spine of interest in order to calculate the local dendritic signal, $\Delta F/F_{0\_dendrite}$. Plotting $\Delta F/F_{0\_spine}$ against $\Delta F/F_{0\_dendrite}$ reveals two components of spine signals, a dendrite-related component and a spine-specific component. The dendrite-related component was removed from the spine signals by subtracting a scaled version of the dendritic signal, $\Delta F/F_{0\_spine\_specific} = \Delta F/F_{0\_spine} - \alpha \, \Delta F/F_{0\_dendrite}$, where $\alpha$ equals the slope of a robust regression and was determined using the MATLAB function 'robustfit' of $\Delta F/F_{0\_spine}$ versus $\Delta F/F_{0\_dendrite}$. To track the activity of the same dendritic spines over many days, individual single time series stacks were first full-frame registered using the motion correction software 'Moco' (*Dubbs et al., 2016*). Stacks were then visually inspected for movement in the z-axis and discarded if a stable dendrite was not evident. Registered stacks were then concatenated across multiple imaging sessions and a running average (100 frames) was generated. The running average stack was then horizontally stitched to the raw stack and used as a template to register the raw data across multiple imaging sessions again using 'Moco' and 'Manual landmark selection' image registration software (ImageJ). Stacks were then visually inspected for active dendrites, the area around the active dendrite was cropped and this sub-section of the original stack was again motion corrected using 'Moco'. Dendritic sections were excluded if there was evidence of crossing axons or dendrites. Calcium responses from individual spines were analyzed using peak detection to determine the onset of a spine response greater than a 15% $\Delta F/F_0$ threshold above baseline (threshold calculated as 3.5× RMS of spine response traces). Spine response frequency was calculated as the number of spine response peaks above threshold per second for the entire imaging session. Spine amplitude was calculated as the average area under the spine response peak which was above threshold, measured across the entire imaging session, except in RSC experiments (*Figure 4*, *Figure 4—figure supplement 1*) where auditory and visual spine response amplitude was measured during auditory and visual stimulation, respectively (or as noted in figure legends). Spine activity integral was the summed area under the curve of all spine responses during the entire imaging session. All data were analyzed blind to time and experimental condition by two independent analysts and differences of more than 10% in responses were investigated. Spines were considered active during a time point if they had at least one response above threshold. Spines were considered inactive during a time point if they had no responses above threshold. Persistent spines were active at all time points. All included spines were active during at least one time point. The orientation selectivity index was calculated as in previous work for cellular responses (*Barnes et al., 2015*; *Ko et al., 2011*). Global dendritic signals were measured for each spine in the dendritic area adjacent to that spine. This approach avoids the ambiguity associated with determining whether or not dendritic branches are contiguous when they enter and exit the imaging depth plane.

## Functional spine classification

To define functional properties of spines in visual cortex, we first classified spines as visual or not based on their responses to visual gratings and a sparse noise stimulus. Spines were determined to be visually responsive if they responded to either set of visual stimuli. First, to determine if a spine responds to the sparse noise stimulus, we used analyses as described previously (*de Vries et al., 2020*). Briefly, for each frame of the stimulus, we calculated the $\Delta F/F_0$ for 0.47 s after the frame (spanning 7 frames) where $F_0$ is the response of the spine in the preceding 1.0 s (15 frames). As had been previously described, we calculated the derivative of the $\Delta F/F_0$ signal immediately after the stimulus presentation and then again 100 ms later and fit a two-dimensional Gaussian-noise distribution, where outliers outside of four standard deviations were considered responsive trials (*de Vries et al., 2020*).

We next determined if spines were responsive to the visual gratings (*Barnes et al., 2015*; *Ko et al., 2011*). We first computed the correlation of the spine calcium fluorescence signal with a binarized visual stimulation signal that corresponds to the timing of the stimulus. In cases where there was a

significant positive correlation with this signal, we then used a false positive threshold analysis to determine the extent to which individual spines had responses that were time-locked to the visual stimulus by chance. To set a threshold for visual responsivity, we used epochs of spine $\Delta F/F_0$ signals collected when the mouse was stationary in the dark and a dummy binary visual stimulation signal identical to that used to detect visual spine correlations. Time-locked responses to the stimulus, occurring within 500 ms of the onset of the stimulus (*Barnes et al., 2015*; *Ko et al., 2011*), were measured in the dark with respect to the dummy visual signal for each spine to determine the false positive rate (mean population false positive rate = 5.5 ± 0.1%). These values were then compared to the percentage of actual visual responses measured during grating presentations. All of the spines (100%) that were identified as visually responsive by our initial correlation analysis had a greater number of stimulus-locked visual responses than the false positive threshold. Because we found few spines that did not respond to both visual stimuli, we only used gratings in later experiments, as the sparse noise stimulus takes approximately 1 hr to present.

To classify network-correlated spines, we calculated the correlation coefficient between individual spine $\Delta F/F_0$ signals and a network average $\Delta F/F_0$ signal comprising of the average of all spine signals during all conditions in the imaged cortical region, excluding the spine of interest (based on *Okun et al., 2015*). Spine $\Delta F/F_0$ signals were determined to be correlated with the network signal if the correlation value was positive and significant (p<0.05). Spines whose activity had positive and significant correlations with the network activity, but not the visual stimulus, were classified as network-correlated spines. Spines were classified as visually responsive using the criteria above, even though their activity also had significant correlations with the network activity (all visually responsive spines also had positive and significant network correlations). Spines whose activity had no significant correlations with either network or visual signals were labelled unclassified.

Using the same visual stimulation threshold criteria mentioned above, 99% of the spines that were identified as network-correlated, but not visually responsive, had fewer stimulus-locked visual responses than our false positive threshold, and non-significant visual correlations. The 1% of identified network-correlated spines (10 spines) with stimulus-locked responses above the false positive threshold were excluded from further analysis. We repeated this entire process in RSC, but also included a binary signal corresponding to the presentation of auditory stimuli. Unclassified spines in both brain regions had no significant correlation with the external stimuli or network signal. Few unclassified spines had significant correlations with either whisker movement (3.2%) or pupil position or diameter (1.4%).

To test for spatial clustering of dendritic spine functional types, we calculated the probability of spines 'n' positions away from a spine$_0$ matching in functional property (adapted from a previous analysis; *Barnes et al., 2017*). The average probability of a functional match was calculated for spines 1,2,3,…,n spines away from spine$_0$, where a match was given a value of 1 and a non-match a value of 0. We repeated this measure for each spine on a branch serving as spine$_0$. We then randomly shuffled spine positions, repeated our analysis to generate a shuffled condition for each branch and compared the shuffled distribution to the experimentally observed distribution.

For within spine comparisons of normalized frequency and amplitude (*Figure 1—figure supplement 1F*, *Figure 3G–I*), we compared frequency at 12 hr and amplitude at 48 hr, since a decrease in synaptic frequency preceded an increase in the strength in past work (*Lee et al., 2010*).

## Statistics

Statistical analyses were performed in either MATLAB or SigmaPlot. Data were tested for equal variance and normality (Shapiro-Wilk test) and then comparisons were made using parametric or non-parametric tests, as appropriate: Student's t-test, paired Student's t-test, chi-square test, one-way ANOVA with Holm-Sidak post hoc test, repeated measures ANOVA with Holm-Sidak post hoc test or a two-way ANOVA with Holm-Sidak post hoc test. Statistical tests were two-sided. Correlation coefficients were calculated with a Pearson's correlation coefficient or a Spearman's rank correlation. A power analysis was performed to ensure we used a sufficient sample size. Specific statistical tests used for all figures along with the number of samples can be found in the source data files.

## Data and code availability

The pre-processed raw data can be accessed at https://doi.org/10.5281/zenodo.7399601. Data that has not been pre-processed is available upon request to any interested party, due to size constraints, by emailing georg.keller@fmi.ch, who will provide temporary transfer access for downloading the data. No proposal is required to access the data and there are no restrictions on who can access the data. Software for controlling the two-photon microscope and pre-processing of the calcium imaging data is available on https://sourceforge.net/projects/iris-scanning/.

## Acknowledgements

We thank Claire Cheetham, Claudia Clopath, Matthias Heindorf, Nicholas A Lesica, and Tobias Rose for comments on the manuscript and helpful discussions. This work was supported by the Edmond J Safra Foundation (SJB), the European Research Council (TK (homeostasis_in_vivo), GBK), the Royal Society (TK), the Wellcome Trust (TK, 212264/Z/18/Z), the Novartis Research Foundation (GBK), the Swiss National Science Foundation (GBK), and the UK Dementia Research Institute at Imperial College London (SJB), which receives its funding from DRI Ltd, funded by the UK Medical Research Council, Alzheimer's Society, and Alzheimer's Research UK. XPRo1595 was provided by Xencor, Inc.

## Additional information

### Funding

| Funder | Grant reference number | Author |
| --- | --- | --- |
| Wellcome Trust | 212264/Z/18/Z | Tara Keck |
| European Research Council | homeostasis_in_vivo | Tara Keck<br>Georg B Keller |
| Royal Society | Wolfson Research Award | Tara Keck |
| Novartis Stiftung für Medizinisch-Biologische Forschung | | Georg B Keller |
| Edmond J. Safra Philanthropic Foundation | | Samuel J Barnes |
| Imperial College London | | Samuel J Barnes |

The funders had no role in study design, data collection and interpretation, or the decision to submit the work for publication. For the purpose of Open Access, the authors have applied a CC BY public copyright license to any Author Accepted Manuscript version arising from this submission.

### Author contributions

Samuel J Barnes, Conceptualization, Data curation, Formal analysis, Methodology, Writing - original draft, Writing - review and editing; Georg B Keller, Resources, Software, Formal analysis, Funding acquisition, Investigation, Methodology, Writing - review and editing; Tara Keck, Conceptualization, Data curation, Formal analysis, Supervision, Funding acquisition, Investigation, Methodology, Writing - original draft, Project administration, Writing - review and editing

### Author ORCIDs

Georg B Keller (ID) http://orcid.org/0000-0002-1401-0117
Tara Keck (ID) http://orcid.org/0000-0001-6623-1037

### Ethics

Experiments were conducted in strict accordance with the United Kingdom Animals (Scientific Procedures) Act 1986, and were approved by the UCL Animal Welfare and Ethical Review Body (AWERB) and by the Veterinary Department of the Canton of Basel-Stadt, Switzerland.

Decision letter and Author response

Decision letter https://doi.org/10.7554/eLife.81958.sa1
Author response https://doi.org/10.7554/eLife.81958.sa2

## Additional files

### Supplementary files
• MDAR checklist

### Data availability

The pre-processed raw data can be accessed at https://doi.org/10.5281/zenodo.7399601. Data that have not been pre-processed are available upon request to any interested party, due to size constraints, by emailing georg.keller@fmi.ch, who will provide temporary transfer access for downloading the data. No proposal is required to access the data and there are not restrictions on who can access the data. Software for controlling the two-photon microscope and pre-processing of the calcium imaging data is available on https://sourceforge.net/projects/iris-scanning/ (copies archived at swh:1:rev:1f1b5616aaaf6cf862d2f4b467b65b479a2b0416, swh:1:rev:48e0b534668ef3ecb7c-2580c00cdbcf650d69043, and swh:1:rev:d3b0d1a0482b5b929621f7e6e99c0e9ff2eee11f).

The following dataset was generated:

| Author(s) | Year | Dataset title | Dataset URL | Database and Identifier |
|---|---|---|---|---|
| Barnes SJ, Keller GB, Keck T | 2022 | Homeostatic regulation through strengthening of neuronal network correlated synaptic inputs | https://doi.org/10.5281/zenodo.7399601 | Zenodo, 10.5281/zenodo.7399601 |

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
