## [Editor Report]

When sensory inputs, such as vision or sound, are chronically disabled, the loss of input activity is counterbalanced by the upregulation of synaptic activity. In this study, the authors have addressed homeostatic responses in adult animals and provide evidence that instead of synapses that directly represent the sensory information, synapses that show correlated intrinsic network activity are the ones that undergo the change upon sensory deprivation. This fundamental and important paper whose key claims are well supported by the data, will be useful to readers in the fields of experience-dependent plasticity, sensory cortical coding, and homeostatic plasticity.

---

## [Decision Letter]

**Decision letter after peer review:**

Thank you for submitting your article "Homeostatic regulation through the strengthening of neuronal network-correlated synaptic inputs" for consideration by *eLife*. Your article has been reviewed by 3 peer reviewers, and the evaluation has been overseen by a Reviewing Editor and Lu Chen as the Senior Editor. The following individuals involved in the review of your submission have agreed to reveal their identity: Yukiko Goda (Reviewer #1); Keith B Hengen (Reviewer #3).

Essential revisions:

All three reviewers have found the work to be important and of considerable general interest. Please carefully address individual points raised by the three reviewers, which can be dealt with by text edits or minor figure modifications. In particular, the fact that the present study used adult animals and that homeostatic plasticity in the adult brain could follow different rules from juvenile animals should be stressed.

*Reviewer #1 (Recommendations for the authors):*

1) Figure 2C – It would be important to include also a histogram of % visual stimuli eliciting time-locked responses for all deprived spines following deprivation as a measure of how well the system is responding to deprivation. In other words, is there an effect of enucleation on the reliability with which visual stimuli trigger responses?

2) Figure 2 – Related to the point above, only panel B appears to show results pertaining to post-enucleation, which is quite confusing. One should reconsider the panel order, perhaps moving panels B and C with both plots including the respective control and deprived data.

3) Figure 3C – Some network spines do show loss of activity upon sensory deprivation. Given that the proportion of network spines is 65% compared to 24% of visual spines, to begin with, one would expect that increase in the proportion of inactive spines by 10% for network spines (Figure 3F) represents a larger total number of spines than 20% increase for visual spines (Figure 3D). Could the increase in normalized amplitude shown by network spines be a consequence of the reduced activity of some of the network spines and secondary to the direct effects of loss of visual inputs? This point requires consideration/discussion. It would be informative to include a blind assessment of relative proportions of visual, network, and unclassified spines following enucleation and a comparison of the numbers to those of baseline in a graphical representation.

*Reviewer #2 (Recommendations for the authors):*

1. Methods: "Intrinsic signal imaging before enucleation was used to localize the visual cortex…" Were the recorded spines/dendrites in binocular or monocular V1? Baseline visual stimulation (gratings) was bilateral? It seems surprising that each spine's ocular dominance was not determined at baseline – please explain. I would expect different reactions to enucleation in a neuron that was mainly driven by the missing eye vs the intact eye.

2. In the example trace Figure S1B, please label all spine events that were counted for the frequency analysis (Figure S1E).

3. Figure 1E Are these spine calcium transients' responses to a visual stimulus (aligned to stimulation time) or are they 'spontaneous' events aligned by peak or max. slope? If the latter, please don't call them responses but find a more neutral term (e.g. EPSCaTs).

4. Figure 2B: "When we examined the functional strengthening of the maintained visually responsive inputs, we found an inverse relationship, such that the spines with the lowest visual responsiveness prior to deprivation underwent the strongest functional strengthening after deprivation." Why is there a data point at zero visual responses, wouldn't that be an unclassified or network spine?

5. Figure 3 G-I: It is not well explained why you show this lack of correlation and what can be concluded from it. To me, frequency changes point to presynaptic plasticity (either release at the observed synapse or spike frequency of the presynaptic neuron), and amplitude changes to a postsynaptic mechanism. It seems you have both, but rarely in the same spines. I don't like the binning – why not show a cloud of all spines and the correlation coefficient? Try drawing a conclusion from these plots.

6. Figure 4 B: Please show raw traces under conditions of TNF α block. Was the amplitude of Ca transients at baseline affected by the inhibitor? It is my understanding that you use the blocker mainly as a diagnostic criterion for "true homeostatic plasticity", as you draw no further conclusions (e.g. about the potential involvement of glia cells).

7. Figure 4J: How many branches were analyzed? (N missing in Table S4).

8. L.337: "Further work is required to better understand the key neural circuit and molecular features that allow TNF-α to selectively modulate homeostatic adaption and even more specifically, the network correlated spines." You see the enhanced activity of network-correlated spines, but homeostatic plasticity could well happen elsewhere, e.g. in L4. It cannot be concluded that TNF-α selectively acts on the spines you happen to look at (a.k.a. streetlight effect). It would be good to spell out that the locus of homeostatic plasticity could be upstream of the synapses under scrutiny (unless you have good arguments that plasticity is local).

*Reviewer #3 (Recommendations for the authors):*

1. Do the authors have the capacity to separate V1m and V1b in their imaging sessions? The responses to monocular deprivation (via enucleation) in V1m should be purely homeostatic, while the effect in V1b is most likely a combination of homeostatic and Hebbian. The TNF α experiments suggest that a significant component of the data is homeostatic (assuming all TNF α plasticity is homeostatic), but it would be powerful to show that the key effects are evident in V1m. This is certainly not necessary for the paper, but if it were possible, it would be quite nice to see and would complement the TNF experiments.

2. Line 64: change "Following deprivation" to "Following sensory deprivation". At this point, there's been no definition of deprivation so readers from outside the field might be confused.

3. Figure 4: the yellow and gray bars (e.g., 4B) are very confusing – my first impression was that they indicated normal vision and 48h dark deprivation. Then, in F through I, I read the figure as if animals were simultaneously deprived of vision and auditory input (note that there is both a gray bar and plugged ears), but the legend says, "timeline for dark exposure (B) OR auditory ear plug (F)", which seems to suggest that these were not performed concurrently.

4. The authors cite Barnes 2017 and Bridi 2018 as the basis of the hypothesis that subsets of spines are subject to compensatory changes. Barnes 2017 clearly shows this, as well as being in adult animals. Bridi 2018 is based on CP animals and, as far as I can tell, doesn't make any claims about synaptic specificity: they measured extracellular spikes and mEPSCs (in whole cell patch), which cannot resolve synaptic subsets. Their key finding (to my eye) is a novel mechanism for compensatory plasticity, not synaptic specificity:

"In sum, our manipulations of visual input combined with manipulations of spontaneous neuronal activity reveal that (i) changes in overall neuronal firing rate alone are not sufficient to mediate the lowered modification threshold for LTP/LTD nor the increased mEPSC amplitude observed after visual deprivation, (ii) changes in mEPSC amplitude induced by visual deprivation reflect Hebbian potentiation of excitatory synapses enabled by the reduced synaptic modification threshold, and (iii) changes in mEPSC amplitude induced by extreme reductions in neuronal activity reflect NMDAR-independent synaptic scaling."

---

## [Author Response]

Essential revisions:All three reviewers have found the work to be important and of considerable general interest. Please carefully address individual points raised by the three reviewers, which can be dealt with by text edits or minor figure modifications. In particular, the fact that the present study used adult animals and that homeostatic plasticity in the adult brain could follow different rules from juvenile animals should be stressed.

We thank the reviewers for their positive and helpful comments. We agree that the age of the animals is a key consideration for our results and we have changed the manuscript to emphasize this point. We have first made sure that the age of the animals is clear throughout the manuscript (abstract, introduction, results, and discussion). We have also highlighted the specifics of adult plasticity, and its relation to juvenile plasticity, in the introduction and discussion to provide context for our findings, as well as how our work is complementary with previous studies in adult animals.

We have addressed the comments of the individual reviewers below.

Reviewer #1 (Recommendations for the authors):1) Figure 2C – It would be important to include also a histogram of % visual stimuli eliciting time-locked responses for all deprived spines following deprivation as a measure of how well the system is responding to deprivation. In other words, is there an effect of enucleation on the reliability with which visual stimuli trigger responses?

We have double checked and there are no visually-evoked responses following deprivation. It is important to clarify that the visual cortex experiments in our study were done in monocular visual cortex, which we have now specified throughout the manuscript. Because all visual spines also have significant network signals (Figure 2), the spines that were identified as visually responsive during baseline (prior to sensory deprivation), only have network responses following visual deprivation, since there is no longer visual input from the contralateral eye. None of the spines have visually-evoked responses above chance after deprivation. We have specified this in the text now. Lines:

“It is important to note that following sensory deprivation, we no longer observed responses to visual stimuli in the visually responsive spines, since the feedforward input had been permanently removed. The responses of the previously categorized visually responsive spines consisted of network responses following deprivation.”

2) Figure 2 – Related to the point above, only panel B appears to show results pertaining to post-enucleation, which is quite confusing. One should reconsider the panel order, perhaps moving panels B and C with both plots including the respective control and deprived data.

We have made this adjustment and switched the order of the panels 2B and 2C, which we agree makes the manuscript easier to follow.

3) Figure 3C – Some network spines do show loss of activity upon sensory deprivation. Given that the proportion of network spines is 65% compared to 24% of visual spines, to begin with, one would expect that increase in the proportion of inactive spines by 10% for network spines (Figure 3F) represents a larger total number of spines than 20% increase for visual spines (Figure 3D). Could the increase in normalized amplitude shown by network spines be a consequence of the reduced activity of some of the network spines and secondary to the direct effects of loss of visual inputs? This point requires consideration/discussion. It would be informative to include a blind assessment of relative proportions of visual, network, and unclassified spines following enucleation and a comparison of the numbers to those of baseline in a graphical representation.

We thank the reviewer for raising this important point. We have adjusted the text to reflect this point that while the relative fraction of visual spines that become inactive is higher than the relative fraction of inactive network spines, the absolute number of network spines that become inactive is larger. Lines:

“While a larger fraction of visually responsive spines become inactive, the absolute number of network-correlated spines that became inactive was higher, since network-correlated spines reflect a larger proportion of the total spine population. These results suggest that the increase in spine loss previously reported following enucleation (Barnes et al., 2017) may include both spines that were visually responsive and network-correlated prior to deprivation (Figure 3D-F).”

Secondly, the increase in the number of inactive network spines and subsequent strengthening only occurs in deprived animals, so even if the effect is secondary to visual deprivation, our data indicate that it is triggered by deprivation. We consistently see that there are not systematic changes in density/size/functional response strength of spines without the visual deprivation in our previous work (Barnes et al., 2017), as well as in this study.

As for the proportion of spines in each category following enucleation, the spines fall into only two categories, network and unclassified, since we do not observe visually-evoked responses following enucleation, as we are recording in monocular visual cortex. We do not see a difference in the unclassified spines following deprivation (0% of unclassified spines (categorized prior to deprivation) correlate with the network signal after deprivation), so there is no change in categorization, other than the visual spines would now be categorized as network-correlated.

Reviewer #2 (Recommendations for the authors):1. Methods: "Intrinsic signal imaging before enucleation was used to localize the visual cortex…" Were the recorded spines/dendrites in binocular or monocular V1? Baseline visual stimulation (gratings) was bilateral? It seems surprising that each spine's ocular dominance was not determined at baseline – please explain. I would expect different reactions to enucleation in a neuron that was mainly driven by the missing eye vs the intact eye.

We apologize for the confusion on this point and have now clarified in the manuscript that the visual cortex experiments in the study were done in monocular visual cortex. All visual stimulation was bilateral, which we have now clarified in the methods. Given that we were measuring in monocular visual cortex, we did not measure ocular dominance. Furthermore, given that we see no visually-evoked responses to bilateral visual stimulation following sensory deprivation, we think that it is unlikely that spines are responding to inputs from the intact ipsilateral eye, consistent with expectations in monocular visual cortex.

2. In the example trace Figure S1B, please label all spine events that were counted for the frequency analysis (Figure S1E).

We thank the reviewer for this helpful suggestion and have denoted events that were included in our analysis with an asterisk above the trace.

3. Figure 1E Are these spine calcium transients' responses to a visual stimulus (aligned to stimulation time) or are they 'spontaneous' events aligned by peak or max. slope? If the latter, please don't call them responses but find a more neutral term (e.g. EPSCaTs).

This is a great point and we have changed the figure to refer to them as spine events. These examples include all spine events, which were aligned by the peak. We have clarified this in the figure legend.

4. Figure 2B: "When we examined the functional strengthening of the maintained visually-responsive inputs, we found an inverse relationship, such that the spines with the lowest visual responsiveness prior to deprivation underwent the strongest functional strengthening after deprivation." Why is there a data point at zero visual responses, wouldn't that be an unclassified or network spine?

We thank the reviewer for catching this mistake. This figure includes all spines, not just the visually responsive ones, and we have changed this text accordingly.

5. Figure 3 G-I: It is not well explained why you show this lack of correlation and what can be concluded from it. To me, frequency changes point to presynaptic plasticity (either release at the observed synapse or spike frequency of the presynaptic neuron), and amplitude changes to a postsynaptic mechanism. It seems you have both, but rarely in the same spines. I don't like the binning – why not show a cloud of all spines and the correlation coefficient? Try drawing a conclusion from these plots.

We have included this comparison, as previous work has indicated that frequency and amplitude could be modulated within a single synapse (Lee et al., 2010). In other words, the frequency of responses in a spine goes down (likely presynaptically mediated) and then the response amplitude of that spine increases. The correlation we are measuring here tests that directly and finds that decreases in frequency in an individual spine is unlikely to be driving the increases in amplitude. We have included this conclusion in the manuscript now. Lines:

“Previous work has shown that spine specific decreases in response frequency drive synapse-specific increases in response amplitude (Lee et al., 2010). We therefore examined if frequency changes were indicative of amplitude changes in individual spines. We observed no correlation between changes in frequency and amplitude within individual spines (Figure 1 – —figure supplement 1F), indicating that the changes in frequency are not likely to be driving the changes in amplitude within a single spine.”

6. Figure 4 B: Please show raw traces under conditions of TNF α block. Was the amplitude of Ca transients at baseline affected by the inhibitor? It is my understanding that you use the blocker mainly as a diagnostic criterion for "true homeostatic plasticity", as you draw no further conclusions (e.g. about the potential involvement of glia cells).

We have added these traces to Figure 4B. The baseline amplitude of Ca transients is not altered by the TNF-α inhibitor (TNF-α 79.2 ± 4.2 vs Control 81.8 ± 3.1 % ∆F/F, p>0.5, t-test). As the reviewer suggests, we have used this experiment to determine whether or not our effect is dependent on TNF-α, which is often used as an indicator of synaptic scaling. It is still unclear exactly how TNF-α interacts with synaptic strengthening, but we have included the known role of glia in this interaction in the discussion.

7. Figure 4J: How many branches were analyzed? (N missing in Table S4).

In order to measure the global dendritic signals, we measured the dendrite signal adjacent to the dendritic spine. Thus, the number of spine and dendrite measurements are the same. This particular approach is critical to avoid any ambiguity that occurs in identifying contiguous dendritic branches when a dendrite exits and re-enters the imaging plane. We have clarified this point in the methods. Lines:

“Global dendritic signals were measured for each spine in the dendritic area adjacent to that spine. This approach avoids the ambiguity associated with determining whether or not dendritic branches are contiguous when they enter and exit the imaging depth plane.”

8. L.337: "Further work is required to better understand the key neural circuit and molecular features that allow TNF-α to selectively modulate homeostatic adaption and even more specifically, the network correlated spines." You see the enhanced activity of network-correlated spines, but homeostatic plasticity could well happen elsewhere, e.g. in L4. It cannot be concluded that TNF-α selectively acts on the spines you happen to look at (a.k.a. streetlight effect). It would be good to spell out that the locus of homeostatic plasticity could be upstream of the synapses under scrutiny (unless you have good arguments that plasticity is local).

We agree with the reviewer and have adjusted the language about TNF-α and added text to highlight that the locus of plasticity could be outside of layer 5. Lines:

“Finally, it is important to note that while we made these measurements in layer 5 pyramidal cells, the homeostatic changes mediated by TNF-α could occur outside of layer 5, including changes to upstream inputs or changes to the presynaptic responses, either through changes in presynaptic release (Vitureira et al., 2012) or through a change in activity patterns of the presynaptic cell (e.g., bursts compared to single spikes) (Linden et al., 2009). Given that both sensory responsive spines (which do not strengthen) and network-correlated spines (which do strengthen) exhibit only network-correlated responses after sensory deprivation, changes in presynaptic activity patterns of network signals would require targeted changes to the network-corelated spines, but not the previously sensory responsive spines. Further research is necessary to identify the role of presynaptic plasticity in homeostatic synaptic plasticity in vivo.”

Reviewer #3 (Recommendations for the authors):1. Do the authors have the capacity to separate V1m and V1b in their imaging sessions? The responses to monocular deprivation (via enucleation) in V1m should be purely homeostatic, while the effect in V1b is most likely a combination of homeostatic and Hebbian. The TNF α experiments suggest that a significant component of the data is homeostatic (assuming all TNF α plasticity is homeostatic), but it would be powerful to show that the key effects are evident in V1m. This is certainly not necessary for the paper, but if it were possible, it would be quite nice to see and would complement the TNF experiments.

We again apologize for this confusion. We have now clarified throughout the manuscript that the visual cortex experiments in our study were conducted in monocular visual cortex and so all of the TNF-α effects in visual cortex are in monocular visual cortex.

2. Line 64: change "Following deprivation" to "Following sensory deprivation". At this point, there's been no definition of deprivation so readers from outside the field might be confused.

We have changed this as suggested.

3. Figure 4: the yellow and gray bars (e.g., 4B) are very confusing – my first impression was that they indicated normal vision and 48h dark deprivation. Then, in F through I, I read the figure as if animals were simultaneously deprived of vision and auditory input (note that there is both a gray bar and plugged ears), but the legend says, "timeline for dark exposure (B) OR auditory ear plug (F)", which seems to suggest that these were not performed concurrently.

We thank the reviewer for pointing out this confusing issue. The dark exposure and auditory ear plug were not performed concurrently, but are separate sets of experiments. We have changed Figure 4 and Figure 4 – —figure supplement 1 so that the background colors are different for the two experiments. We have also clarified in the text that these are separate experiments.

4. The authors cite Barnes 2017 and Bridi 2018 as the basis of the hypothesis that subsets of spines are subject to compensatory changes. Barnes 2017 clearly shows this, as well as being in adult animals. Bridi 2018 is based on CP animals and, as far as I can tell, doesn't make any claims about synaptic specificity: they measured extracellular spikes and mEPSCs (in whole cell patch), which cannot resolve synaptic subsets. Their key finding (to my eye) is a novel mechanism for compensatory plasticity, not synaptic specificity:"In sum, our manipulations of visual input combined with manipulations of spontaneous neuronal activity reveal that (i) changes in overall neuronal firing rate alone are not sufficient to mediate the lowered modification threshold for LTP/LTD nor the increased mEPSC amplitude observed after visual deprivation, (ii) changes in mEPSC amplitude induced by visual deprivation reflect Hebbian potentiation of excitatory synapses enabled by the reduced synaptic modification threshold, and (iii) changes in mEPSC amplitude induced by extreme reductions in neuronal activity reflect NMDAR-independent synaptic scaling."

We thank the reviewer for this helpful point and we have taken this point out of the manuscript.